# Compression for Better: A General and Loss-Driven Compression Framework

## Abstract

This work focuses on general and loss-value-driven lossless model compression, ensuring that the model's loss value remains unchanged or decreases after compression. A key challenge is effectively leveraging compression errors and defining the boundaries for lossless compression to minimize model loss. i.e., compression for better. Currently, there is no systematic approach to determining this error boundary or understanding its specific impact on model performance. We propose a general and loss-driven **L**oss**L**ess **C**ompression theoretical framework (**LLC**), which further delineates the compression neighborhood and higher-order analysis boundaries through the total differential, thereby specifying the error range within which a model can be compressed without loss. To verify the effectiveness of LLC, we apply various compression techniques, including quantization and decomposition. Specifically, for quantization, we reformulate the classic quantization search problem as a grouped knapsack problem within the lossless neighborhood, achieving lossless quantization while improving computational efficiency. For decomposition, LLC addresses the approximation problem under low-rank constraints, automatically determining the rank for each layer and producing lossless low-rank models. We conduct extensive experiments on multiple neural network architectures on different datasets. The results show that without fancy tricks, LLC can effectively achieve lossless model compression. Our code will be made publicly.

## 1. Introduction

The scale and complexity of Deep Neural Networks (DNNs) have rapidly increased, driving up memory and FLOPS demands. To tackle these challenges, model compression has become a crucial method for improving efficiency, reducing energy consumption, and speeding up inference. However, achieving effective compression without sacrificing performance remains a key challenge. As such, model compression must balance two critical objectives: maximizing the compression ratio while preserving model performance. In this work, we focus on post-training compression, where the model is compressed after training without requiring modifications to the training process. Its advantage is that the computing resource consumption is low and there is no need to adjust the model training process.

In the context of post-training compression, developing a framework that preserves model performance while ensuring generality requires two key conditions: **Performance Assurance**, which involves understanding the relationship between compression-induced errors and model performance to maintain accuracy and stability; and **General Applicability**, ensuring the method can be broadly applied across different tasks and technical frameworks.

For the performance assurance, most existing compression schemes focus on maximizing the compression rate while trying to optimize the performance of the compressed model. Taking model quantization and matrix decomposition as examples, quantization significantly speeds up inference and reduces model size by using lower bit widths to represent tensors. For example, HAWQ (Dong et al., 2019) employed layer-wise sensitivity metrics to determine the precision of different layers, striking a good balance between error and compression ratio. Matrix decomposition decomposes the weight matrix into two or more smaller matrices, using these smaller matrices during actual storage and computation. Hsu et al. (Hsu et al., 2022) incorporated weighted Fisher information into singular value decomposition error to reduce the model degradation after decomposition. Although these methods aim to reduce the impact of compression errors at different compression rates, performance degradation remains unavoidable at both high and low compression rates. This is primarily because the optimization of compression errors is not inherently aligned with the optimization of model performance.

For the general applicability, The key is identifying analytical tools or evaluation metrics that are applicable across tasks while remaining independent of specific models or tasks. While task-specific metrics like accuracy and perplexity provide an intuitive measure of model performance, their strong dependence on specific tasks limits their applicability across domains. In contrast, loss functions serve as a universal optimization objective in machine learning, offering

two key advantages: (1) they provide a consistent evaluation standard across tasks while maintaining strong correlations with downstream metrics, and (2) their continuous and differentiable nature enables precise quantification of model variations, offering a solid mathematical foundation for optimization. Building on this, the proposed LLC framework centers on loss functions to analyze the relationship between compression error and model performance, ensuring loss remains unchanged or decreases during compression, thereby enabling domain-independent compression.

LLC reveals the relationship between compression error and model performance through full differential analysis, and uniformly analyzes compression error and model loss. LLC also clearly defines the error neighborhood of lossless compression and determines the boundary through second-order Hessian analysis. We apply model quantization and matrix decomposition to the LLC framework: for quantization, we transform the quantization search problem into a grouped knapsack problem to improve computational efficiency while ensuring lossless quantization; for decomposition, we combine the compression error neighborhood and low-rank constraints to generate lossless low-rank models. Experiments show that LLC can effectively compress models without loss under multiple datasets and different network architectures while ensuring compression rate, and even obtains compressed models with lower loss than the original model. For example, LLC compresses the volume of the ResNet series model by nearly 70%, achieving better performance than the original model.

Our contributions are as follows: 1) We propose a universal loss-driven compression framework that provides guidance on how compression errors can be used for lossless model compression. 2) We apply LLC to quantization and matrix decomposition: by transforming the quantization search problem into a knapsack problem, we ensure lossless compression; in matrix decomposition, we combine the error neighborhood with low-rank constraints to successfully generate lossless low-rank models. 3) Experimental results across multiple task datasets, neural network architectures, and multiple compression technologies verify the effectiveness of the proposed LLC framework.

## 2. Related Works

### 2.1. Quantization

Quantization uses low-bit-width representations for tensors while maintaining their dense format, aiming to reduce model storage and computational overhead. In typical setups, mixed-precision quantization strategies are employed, where different layers are assigned varying bit-widths based on their sensitivity to quantization. This approach minimizes performance loss after compression. For example,

HAQ (Wang et al., 2019) used reinforcement learning to determine the quantization strategy for each layer, incorporating feedback from hardware accelerators to improve computational efficiency. AutoQ (Lou et al., 2019) introduced a layered deep reinforcement learning (DRL) method that sequentially determines kernel bit-widths. HAWQ (Dong et al., 2019) employed the top Hessian eigenvalues to measure each layer's sensitivity to quantization, providing a relative sensitivity score, although bit-width allocation still relies on manual selection. HAWQ-V2 (Dong et al., 2020) replaced this with the trace of the Hessian matrix. BRECQ (Li et al., 2021) further introduced block-wise optimization, which used different granularities of quantization to significantly reduce the model degradation induced by quantization. While these methods narrow the performance gap between the compressed and original models in practice, model degradation is still difficult to fully avoid, even under 8-bit quantization. Moreover, although these methods are effective empirically, they lack a principled explanation of optimality. Furthermore, bit-width assignment for each layer leads to an exponentially growing search space, decreasing efficiency.

### 2.2. Decomposition

Traditional decomposition methods, such as Singular Value Decomposition(SVD), CANDECOMP/PARAFAC(CP), and Tucker decomposition, involve decomposing model weight matrices and directly assigning the decomposed weights back to the original model. However, this approach often leads to significant increases in model loss, typically rising 5–10 times compared to the original model. To mitigate this issue, existing methods incorporate fine-tuning after decomposition, which entails retraining to reduce the loss. Yu et al. (Yu et al., 2017) leveraged weight structure information by combining low-rank weight matrices and feature map reconstruction to reduce fully-connected layer parameters. Xu et al. (Xu et al., 2019) integrated low-rank approximation with regularization into the training process, achieving a notable reduction in performance degradation. Yang et al. (Yang et al., 2020) introduced an SVD-based decomposition training method that first decomposes each layer into full-rank forms and then retrains the decomposed weights. Zhang et al. (Zhang et al., 2023) used multiple low-rank matrices to approximate gated recurrent unit (GRU) weight matrices and subsequently retrained the model. While these methods can mitigate loss through fine-tuning, they still often yield some level of model degradation and entail significant time costs in the retraining phase.

The above methods aim to reduce the gap between the compressed and original models. In contrast to the view that compression inevitably leads to degradation, we aim to offer a method where model loss consistently decreases after compression, without requiring fine-tuning or other additional

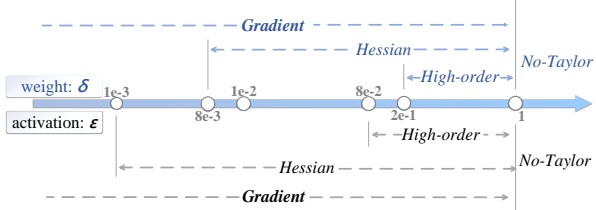

Figure 1. In the analysis of noise boundaries for weights and activations, for activations, when the noise level is below $1 \times 10^{-3}$, only the first-order term needs to be considered, as higher-order terms have negligible impact on optimization. When the noise is below $8 \times 10^{-2}$, the second-order Hessian term should be incorporated along with the first-order gradient term in the optimization objective. For weights, although theoretically a well-trained model should yield zero weight gradients, in practice, the weight gradients are seldom exactly zero and therefore still need to be taken into account.

steps—in other words, compression yields gains.

## 3. Lossless Theoretical Framework

**Basic Analysis.** The theoretical framework of LLC is primarily based on the mathematical properties of extreme points, oriented to the loss function, and aims to reduce the loss value and improve the model performance. In general, for an n-layer neural network model, the loss of the model is optimized according to the following equation

$$\min_{W} f(W) = \mathbf{E}_{Sample} \ell(W, Sample) = \frac{1}{m} \sum_{(x_i, y_i) \in \mathbb{D}} \ell(W, x_i, y_i)$$

$$\ell(W, x_i, y_i) = L(model_n(x_i, W), y_i),$$
$$model_n = h_1(h_2(h_3(h_4(\cdots(h_{n+1}, w_n)\cdots, w_4), w_3), w_2), w_1)$$
(1)

where $f(\cdot)$ represents the loss of the model on a dataset, $\mathbf{E}$ stands for expectation, $m$ is the size of the dataset, $\ell(\cdot)$ is the loss function for a sample, and $(x_i, y_i)$ denotes a sample in the dataset along with its corresponding label, $L(\cdot)$ represents the loss function, such as the cross-entropy function; $h_i$, with $i \in [1, ..., n]$, represents the $(n-i+1)$th layer in the neural network; $W = (w_n^T, w_{n-1}^T, \cdots, w_1^T)^T$, where $w_i$ is the parameter in $h_i(\cdot)$; and for the reason of a unified format, $h_{n+1}$ denotes the sample $x$. This form ensures that LLC is independent of the specific network architecture, making it applicable to different models.

Compression techniques such as quantization and decomposition are mathematically considered to be the process of adding noise to the original weights and activations of the model. After compression, for a sample, the model loss $\ell$ during the inference process is restated as

$$\bar{\ell}(w, x_i, y_i) = L(h_1(h_2(\cdots h_n(h_{n+1} + \epsilon_n, w_n + \delta_n)$$
$$+ \epsilon_{n-1}\cdots, w_2 + \delta_2) + \epsilon_1, w_1 + \delta_1), y_i)$$
(2)

where $\delta_i$, $i \in 1, \cdots, n$, and $\epsilon_i$, $i \in [1, ..., n]$ are errors caused by compression, such as data type conversion in quantization and low-rank error in decomposition.

LLC directly associates the compression noise error and the change of the loss function through total differentials. According to total differentials, the following equation can be obtained

$$\bar{\ell}(w, x_i, y_i) - \ell(w, x_i, y_i) = \sum_{i=1}^{n} \frac{\partial \ell}{\partial h_{i+1}} \cdot \epsilon_i + \frac{\partial \ell}{\partial w_i} \cdot \delta_i +$$
$$\frac{1}{2}(\epsilon_i, \delta_i)^T \mathbb{H}(\epsilon_i, \delta_i) + O(||(\epsilon_i, \delta_i)||^n)$$
(3)

where $\mathbb{H}$ represents the Hessian matrix and $O(||(\epsilon_i, \delta_i)||^n)$ represents the high-order term, $\cdot$ is inner product and $*$ is the scalar product. For the loss on whole dataset, we can gain

$$\min_{\epsilon \in E} \bar{f}(w) - f(w) = \frac{1}{m} \sum_{(x_j, y_j) \in \mathbb{D}} \sum_{i=1}^{n} \frac{\partial \ell}{\partial h_{i+1}} \cdot \epsilon_i + \frac{\partial \ell}{\partial w_i} \cdot \delta_i$$
$$+ \frac{1}{2}(\epsilon_i, \delta_i)^T \mathbb{H}(\epsilon_i, \delta_i) + O(||(\epsilon_i, \delta_i)||^n)$$
(4)

where $\bar{f}(w) = \frac{1}{m} \sum \bar{\ell}(\cdot)$. This equation directly links compression and model performance (Loss). Thus, we can optimize the above expression to make $\bar{f}(w) - f(w) < 0$, meaning that the loss after compression is smaller than the original model's loss.

**Lemma 3.1.** *The total differential describes the increment of a smooth, differentiable function under arbitrarily small parameter changes.*

Lemma 3.1 (Parr & Howard, 2018) sets limitations on the use of the total differential: first, the function must be smooth and differentiable, and second, parameter changes must be sufficiently small. According to the chain rule, multilayer neural networks are continuously differentiable with respect to all parameters, meaning they are inherently smooth and differentiable. Therefore, Eq. 4 generally satisfies $C^k$ continuity. As the scale of compression governs the parameter changes, we primarily focus on the magnitude of noise.

**Lemma 3.2.** *The total differential relies on a linear approximation assumption, valid only when the changes in the function's variables are sufficiently small.*

When the variations $\epsilon, \delta$ are small enough, the actual change in the loss function can be accurately described by the total differential $df$. The lemma above outlines the theoretical range for noise. Thus, it is essential to identify this "sufficiently small" threshold within the practical model.

**Noise Neighborhood Mapping.** Since each layer can accommodate different noise sizes, we set the noise to $\delta_i, \epsilon_i$. We then calculate the gap $U(x)$ between theory and practice in Eq. 4

$$U_{\delta^k}(x_i) : |\ell(w \pm \delta_i^k, x_i, y_i) - (\ell(w, x_i, y_i) + \sum_{i=1}^{n} \frac{\partial \ell_k}{\partial w_i} \cdot \delta_i^k +$$
$$\frac{1}{2}(\delta_i)^T \mathbb{H}(\delta_i) + O(||(\delta_i)||^n)|$$
(5)

$$U_{\epsilon^k}(x_i) : |\hat{\ell}(w, x_i, y_i) - (\ell(w, x_i, y_i) + \sum_{i=1}^{n} \frac{\partial \ell_k}{\partial h_{i+1}} \cdot \epsilon_i^k) +$$

$$\frac{1}{2}(\epsilon_i)^T \mathbb{H}(\epsilon_i) + O(||(\epsilon_i)||^n)| \quad (6)$$

the left side represents the loss due to actual noise disturbances, while the right side represents the theoretical loss induced by noise. The $k$ controls the compression level, such as 4/8-bit or rank. When $k$ represents rank, higher values of k result in lower compression. Eq. 5 and Eq. 6 calculate the noise bounds for weights and activations, respectively. LLC measures the change in loss through perturbation of the first- and second-order terms. As shown in Fig. 1, the perturbation noise boundaries under LLC and their dominant influencing orders are illustrated. First, for activations, when the noise range is below $10^{-3}$, the first-order term is the dominant factor, and the gradient can be treated as the primary optimization target, with the influence of higher-order terms negligible. This is because higher-order errors decay exponentially compared to lower-order ones. When the noise range is between $[10^{-3}, 8 \times 10^{-2}]$, both the first- and second-order terms significantly affect the loss, thus the second-order Hessian information should be included in the optimization target. For noise levels above $8 \times 10^{-2}$, the negative impact increases, and higher-order terms need to be considered.

For weights, ideally, the first-order term in a trained-well model should be zero. However, in practice, gradients are rarely zero, so they must be included in the optimization. When the noise is less than $8 \times 10^{-3}$, the first-order term should be the main optimization target. When the noise range is between $[8 \times 10^{-3}, 2 \times 10^{-1}]$, the second-order term's influence should be considered. When the noise exceeds $2 \times 10^{-1}$, the influence of weight noise on the loss becomes significant, and higher-order terms should not be omitted. This range defines the noise boundaries and the dominant terms in the compression process.

Multiple experiments show that weights have higher noise tolerance than activations, meaning weights can be deeply compressed, whereas activations cannot. This phenomenon aligns with the consensus that weights are more easily compressed. Since our focus is on stable and efficient lossless compression, experimental results show that the first-order terms dominate across all noise ranges, while the contribution of higher-order terms exponentially decays and has a minimal effect on loss changes. In the quantization process, we evaluated the impact of the second-order terms, and the error loss was found to be below 0.00001. In decomposition, due to the lack of consideration for the full covariance structure of the original data, errors introduced by the decomposition can severely distort the Hessian matrix, leading to incorrect estimates. Furthermore, the computation of

second-order terms is computationally expensive and time-consuming. Therefore, in our analysis, given the dominant effect of the first-order terms and the consideration of time efficiency, LLC omits higher-order terms, as their impact on performance is negligible. Thus, LLC mainly focuses on gradient-driven lossless compression.

**LLC Framework.** The LLC framework is a highly efficient, lossless compression method based on the first-order analysis range. Within the first-order range, the Eq. 4 is updated to $\min_{\epsilon \in E} \bar{f}(w) - f(w) = \frac{1}{m} \sum_{(x_j, y_j) \in \mathbb{D}} \sum_{i=1}^{n} \frac{\partial \ell}{\partial h_{i+1}} \cdot \epsilon_i + \frac{\partial \ell}{\partial w_i} \cdot \delta_i$. We need to find appropriate noise vectors $\epsilon$ and $\delta$ to obtain a model with minimal loss. When the inner product is negative, the compressed model's loss is lower than the full-precision model, meaning we find noise vectors opposite to the model's gradient direction. Thus, in theory, the goal of loss-driven lossless compression is achieved.

To select the appropriate compression noise, we must ensure that it opposes the model gradient direction. Taking activation compression as an example, we will first explain the rationale behind this choice. Using the language of probability theory, we describe $\frac{\partial \ell}{\partial h_{i+1}} \cdot \epsilon$ for $\epsilon$ is a stochastic vector naturally. Let $\epsilon = [e_1, e_2, .., e_k]$ and $e_i$ is i.i.d. random variable. We also set that $\frac{\partial \ell}{\partial h_{i+1}} = [p_1, p_2, ..., p_k]$ and $p_i$ represent i.i.d. random variable. $e$ and $p$ are independence to each other, $k$ is the length of the vector. Alternatively, $p_i$ can be treated as the random variable with different distributions or directly use $\mathbf{E}\frac{\partial \ell}{\partial h_{i+1}}$ vector in analyses. The conclusions are the same or close with current analysis. We have $\frac{\partial \ell}{\partial h_{i+1}} \cdot \epsilon = \sum_{i=1}^{k} e_i * p_i$ and Eq. 7

$$\mathbf{E}\frac{\partial \ell}{\partial h_{i+1}} \cdot \epsilon = \mathbf{E} \sum_{i=1}^{k} e_i * p_i = k\mathbf{E}e\mathbf{E}p \quad (7)$$

For a well-trained model, the $\mathbf{E}p$ can be computed as $\mathbf{E}p = \frac{1}{k} * \frac{\partial \ell}{\partial h_{i+1}} \cdot \vec{1}$. Then to gain a negative $\mathbf{E}\frac{\partial \ell}{\partial h_{i+1}} \cdot \epsilon$, the $\mathbf{E}e$ should be different signs with $\mathbf{E}p$. For specific compression methods, such as quantization, we use different rounding functions to ensure the sign of $\mathbf{E}e$. In decomposition, we calculate the noise direction at different ranks. This type of method is not the only way to obtain a negative inner product, but it is easy to calculate and effective.

After having a compression method, we also need to analyze the performance improvement brought by the compressed model and the probability of obtaining a lower loss model. Based on the above analysis and the Chebyshev's inequality, we can infer

$$P(\frac{\partial \ell}{\partial h_{i+1}} \cdot \epsilon \geq 0) < P(|\frac{\partial \ell}{\partial h_{i+1}} \cdot \epsilon - \mathbf{E}e\mathbf{E}p| \geq |\mathbf{E}e\mathbf{E}p|)$$
$$\leq \frac{Var(ep)}{|\mathbf{E}e\mathbf{E}p|^2} = \frac{Var(e)Var(p)}{|\mathbf{E}e\mathbf{E}p|^2} + \frac{Var(e)}{|\mathbf{E}e|^2} + \frac{Var(p)}{|\mathbf{E}p|^2} \quad (8)$$

Hence, when $\mathbf{E}p$ is larger, i.e., $|\frac{\partial \ell}{\partial h_{i+1}} \cdot \vec{1}|$ is larger, $Var(p)$ is smaller, making it more likely to obtain good results.

## 4. LLC Quantization and Decomposition

**Quantization.** Loss-Driven lossless mixed-precision quantization addresses two key challenges: first, how to achieve stable lossless compression under mixed-precision quantization; and second, how to efficiently select the optimal quantization bit-width for each layer, which is an NP-hard problem. For the first challenge, LLC quantization is applied for first-order analysis, ensuring lossless quantization within the first-order bounds. The second challenge is reformulated as a group knapsack problem, which is solved efficiently using dynamic programming. In the LLC framework, the loss function is treated as the "value $P$", each layer $i$ is considered a "group" with one bit-width $j$ choice per group, and the model size is treated as the "knapsack capacity $W$". This transforms the original problem into a low-computation group knapsack problem, where the goal is to select the optimal bit-width for each layer to minimize loss while keeping the quantized model size within the specified capacity $C$.

$$\min \sum_{i=1}^{n} P[i][j] \quad s.t. \sum_{i=1}^{n} W[i][j] < C, j \in [1, k], j \in \mathbf{Z} \quad (9)$$

where $n$ is the number of model layers. The problem scale of the grouped knapsack is very small, usually less than $n * k$, and has a significant efficiency advantage. The overall process of our proposed method is shown in Algorithm 1. In the algorithm, $\epsilon$ and $\delta$ are the quantization noise errors of activation and weight. Positive and negative are the choices of different quantization directions. We set the quantization level of Algorithm 1 to $k = 4$ categories, namely 2/4/8/16 bit. The total time complexity is $O(n * k * feature)$.

**Decomposition.** The main challenge of post-training decomposition is how to choose a low rank, so as to reduce the model loss stably while compressing. Under the LLC framework, we view the decomposition problem as a numerical rank-deficiency issue and study how the rank of weight matrices at different layers affects the final model loss. In our decomposition approach, we opt for the simplest low-rank decomposition scheme due to its minimal parameter introduction and highest efficiency.

We treat the LLC error calculation boundary as a differential neighborhood and combine it with the low-rank assumption as an inequality constraint in the optimization objective, as shown below

$$\min_{\delta^k \in \Delta} \bar{f}(w) - f(w) = \frac{1}{m} \sum_{i=1}^{n} \sum_{(x_j, y_j) \in \mathbb{D}} \frac{\partial \ell}{\partial w_i} \cdot \delta_i^k \quad (10)$$

$$s.t. \ U_{\delta^k} : \{ \|w_{ij} - l_{ij} r_{ij}\|_F \}_{i,j} \leq \gamma, \ \forall i, j \quad (10a)$$

$$0 < k < \frac{NM}{N + M} \quad (10b)$$

where since the calculated error gamma vector is the smallest, we choose the $F$ norm and approach it to 0. This algorithm is flexible, the neighborhood calculation can be replaced with other decomposition methods, such as

---

**Algorithm 1** Lossless Mixed Precision Search Grouped Knapsack Algorithm

1: **Input:** Neural network $M$ with $n$ layers, quantization levels $[q_1, q_2, ..., q_k]$, maximum error $error_{max}$, calibration dataset $D$
2: **Output:** Cost matrix $P$, weight matrix $W$ of size $n \times k$
3: Calibrate the network $M$ with dataset $D$ to collect data distribution
4: **for** each $q_j$ in $[q_1, q_2, ..., q_k]$ **do**
5:     **for** each $Layer_i$ in $M$ **do**
6:         Calculate $W[i][j]$, the model size of $Layer_i$ at $q_j$
7:         Compute $\|\epsilon_i\|$ and $scale_{input}$ for $Layer_i$
8:         Calculate $slope = \frac{\|f(M) - f_{input}(M; scale_{input}, i)\|}{scale_{input}}$
9:         Compute $fluc$ as
    $\|f(M) - f_{weight}(M; scale_{weight}, i)\|$
10:         Determine noise for quantization:
11:         **if** Positive **then**
12:             $noise = scale_{input} \times \lceil random \rceil$
13:         **else if** Negative **then**
14:             $noise = scale_{input} \times \lfloor random \rfloor$
15:         **end if**
16:         **if** $fluc < error_{max}$ **then**
17:             Update $P[i][j]$ with $slope \times \frac{\|\epsilon_i\|}{\sqrt{size(e_i)}}$
18:         **else**
19:             Calculate $\|\delta_i\|$ and $scale_{weight}$
20:             Update $P[i][j]$ with $slope \times \frac{\|\epsilon_i\|}{\sqrt{size(\epsilon_i)}} + \frac{fluc}{scale_{weight}} \times \frac{\|\delta_i\|}{\sqrt{size(\delta_i)}}$
21:         **end if**
22:     **end for**
23: **end for**
24: **return** $P, W$

---

$\hat{w} = usv^T$. However, using alternative decompositions may increase parameter count and computation time.

Algorithm 2 is our proposed lossless decomposition method. This algorithm incorporates a layer-wise early stopping strategy: during the decomposition process of each layer, if a candidate decomposition meets the loss threshold with a sufficiently low loss, the search for additional candidate matrices for that layer is immediately halted, enhancing efficiency. The time complexity is $\mathcal{O}(n * k * feature)$, where $k << r_{max}$ represents the actual number of decompositions per layer.

## 5. Experiment

### 5.1. Datasets and Details.

**Datasets.** The ImageNet-1K dataset (Krizhevsky et al., 2017) consists of 1.28 million training and 50K valida-

**Algorithm 2** Lossless Decomposition Algorithm under Numerical Rank-Deficiency

---

**Input:** Neural network $M$ with $n$ layers, loss threshold $\epsilon$, maximum rank $rank_{max}$ .

**Output:** Loss-minimized model $\hat{M}$ after factorization.

1: **for** $Layer_a$ in $M$ **do**
2:     **if** $Layer_a$ is already decomposed **then**
3:        continue
4:     **end if**
5:     Initialize loss list $A$ for recording candidate factorizations.
6:     **for** $c = 1, 2, ..., rank_{max}$ **do**      ▷ Parallel optimization
7:        Initialize $L_c, R_c$     ▷ Temporary matrices for rank-$c$ factorization
8:        Compute the error $\delta_i$ of $h_{i+1}$ under rank-$c$ level on the dataset
9:        **if** $\{\|w_{ij} - l_{ij}r_{ij}\|_F\}_{i,j} \leq \gamma, \forall i, j$ **then**
10:           **if** $\frac{\partial \ell}{\partial w_i} \cdot \delta_i < 0$ **then**
11:              {Record $L_c, R_c$, and computed Loss} in list $A$
12:              **if** $\{\|w_{ij} - l_{ij}r_{ij}\|_F\}_{i,j} \rightarrow \mathbf{0})$ **then**
13:                 **break**     ▷ Early stopping
14:              **end if**
15:           **end if**
16:        **end if**
17:        Update $L_c, R_c$       ▷ Update
18:     **end for**
19:     Select $L_a, R_a$ from $A$ that minimizes Loss
20:     $W_a = L_a \cdot R_a$    ▷ Final factorized matrix for layer
21:     Return $\hat{Layer_a}$
22: **end for**
23: Return $\hat{M}$

---

*Table 1.* Activation under different models introduces different levels of compressed noise neighborhoods for first-order terms.

| $\epsilon$ | ResNet-18 | ResNet-34 | ResNet-50 | ResNet-101 | BERT |
|---|---|---|---|---|---|
| [1e-1] | 0.00735 | 0.009541 | 0.023649 | 0.020001 | 0.011155 |
| [8e-2] | 0.005322 | 0.007455 | 0.013232 | 0.006897 | 0.008154 |
| [1e-2] | 0.004321 | 0.006581 | 0.008651 | 0.001548 | 0.005221 |
| [1e-3] | 0.002283 | 0.004321 | 0.006422 | 0.000801 | 0.004517 |
| [1e-4] | 0.002675 | 0.004362 | 0.006458 | 0.000823 | 0.004394 |

error bounds can be flexibly computed using Eq. 5 and Eq. 6 across various models on multiple datasets. Experiments show that, although the error bounds vary, the majority of models fall within this defined range. The parameters $error_{max}$ and $\gamma$ are set to approximately $10^{-4}$ in the algorithm. Quantization parameters are calculated using the ACIQ method. The validation set of ImageNet is used as the calibration set, where we check gradients without updating the weights. To ensure fairness, all experiments are conducted under identical optimization settings and executed on two NVIDIA A800 GPUs. The models are implemented based on pre-trained full-precision configurations in PyTorch. The code is implemented in PyTorch.

### 5.2. Ablation

**Compressed Noise Bounds.** The calculation of error bounds depends on the sensitivity of different models to noise, resulting in varying error bounds for each model. When a model is sensitive to noise, the extent of lossless compression is limited. The error neighborhood extends beyond the analytically manageable range of total differentials, making stable lossless compression unachievable. Firstly, as shown in Table 1, we present the first-order analysis error bounds for different models on ImageNet. The data in the table are the actual calculation results of different models in Eq. 6. When the noise is large, $U_{\epsilon^k}$ will also increase, indicating that there is a gap between the theoretical calculation results and the actual. According to Equation 3, when the error is less than 1, the second-order term is the square of the error, which further diminishes the influence of the second-order term, establishing that the first-order analysis error is dominant. Experimental results indicate that when noise is low, the actual results for LLC align closely with theoretical predictions. The data in the table suggest that the loss impact from the second-order term is negligible. For instance, if we want the impact of the loss function to be less than $6 * 10^{-5}$, which is the minimum positive number for FP16 ($\epsilon < \sqrt{0.00006}$), resulting in a small second-order impact, the first-order derivative estimation performs effectively.

**Weight Gradient and Compression Level.** In theory, the weight gradients of a well-trained model should be close to zero. However, experimental results show that while weight gradients are generally small, they are not precisely zero. Thus, when compression noise is introduced, the impact of weight changes on the loss function is minimal. Compared to activations, weights can tolerate higher compression levels. Based on experiments across various models

tion images. ImageNet-1K is usually used as the benchmark for model compression. SWAG dataset (Zellers et al., 2018) consists of 113k multiple-choice questions about grounded situations. The Stanford Question Answering Dataset (SQuAD) (Rajpurkar et al., 2016) is a collection of question-answer pairs derived from Wikipedia articles. In SQuAD, the correct answers to questions can be any sequence of tokens in the given text. MNLI (Williams et al., 2017) is a dataset for natural language reasoning tasks. Its corpus is a collection of textual implication annotations of sentences through crowdsourcing. The task is to predict whether the premise sentence and the hypothesis sentence are logically compatible (entailment, contradiction, neutral).

**Details.** The LLC scheme does not involve fine-tuning or retraining. We utilize the VGG (Simonyan & Zisserman, 2014), MobileNet (Howard, 2017), ResNet (He et al., 2016) series (including ResNet-18, 34, and 50) to determine the error bounds depicted in Figure 1. In the implementation,

*Table 2.* Performance of different models on image datasets. LLC quantize the model and loss is lower than the original model.

| Model | Top-1 | Top-5 | Loss | Bit-width | Drop-rate |
|---|---|---|---|---|---|
| **MNIST** | | | | | |
| CNN | 97.51 | - | 0.0792 | Full Prec. | |
| Ours | **97.66** | - | **0.0786** | Mix(8/4bit) | ↓73% |
| **CIFAR** | | | | | |
| VGG13 | 73.69 | - | 1.2726 | Full Prec. | |
| Ours | **74.09** | - | **1.2503** | Mix(8/4/2bit) | ↓74% |
| MobileNet | 66.21 | - | 1.5653 | Full Prec. | |
| Ours | **66.59** | - | **1.5631** | Mix(8/4/2bit) | ↓69% |
| ResNet-14 | 86.68 | - | 0.3634 | Full Prec. | |
| Ours | **87.23** | - | **0.3576** | Mix(F/8bit) | ↓56% |
| MobileNet_V2 | 62.44 | - | 1.6358 | Full Prec. | |
| Ours | **62.88** | - | **1.6245** | Mix(8/4/2bit) | ↓71% |
| **ImageNet** | | | | | |
| VGG16 | **71.59** | **91.38** | 1.1454 | Full Prec. | |
| Ours | 71.43 | 90.30 | **1.1337** | Mix(F/8/4bit) | ↓77% |
| MobileNet_V2 | 71.89 | 90.29 | 1.1480 | Full Prec. | |
| Ours | **71.89** | **90.30** | **1.1478** | Mix(8/4/2bit) | ↓71% |
| ResNet-18 | **69.77** | 89.07 | 1.2470 | Full Prec. | |
| Ours | 69.72 | **89.09** | **1.2457** | Mix(F/8/4bit) | ↓73% |
| ResNet-34 | **73.29** | **91.43** | 1.0812 | Full Prec. | |
| Ours | 72.88 | 91.24 | **1.0787** | Mix(F/8/4bit) | ↓62% |
| ResNet-50 | 75.06 | 92.42 | 1.0019 | Full Prec. | |
| Ours | **75.09** | **92.44** | **0.9854** | Mix(F/8/4bit) | ↓66% |
| **SQuAD** | | | | | |
| | **EM** | **F1** | **Loss** | **Bit-width** | **Drop-rate** |
| BERT | 80.49 | 88.15 | 0.4461 | Full Prec. | |
| Ours | **80.51** | 88.15 | **0.4461** | Mix(F/8bit) | ↓45% |

and accounting for different sensitivities among layers, we averaged the noise introduced. For example, in quantization, 2-bit quantization introduces noise at an order of $10^{-1}$, 4-bit quantization introduces noise at approximately $5*10^{-3}$, and 8-bit quantization introduces noise around $5*10^{-4}$. Consequently, 4-bit and 8-bit are the primary compression levels used in the LLC framework.

### 5.3. Performance and General Applicability

In the comparison experiments, we conduct LLC-based lossless quantization tests alongside standard benchmarks. The lossless experiments are compared against uncompressed models, while the comparison benchmarks are tested against existing methods to highlight the versatility of LLC under different architectures, datasets, and tasks. In addition, we also verify LLC on different compression techniques.

**Lossless in Quantization.** As shown in Table 2, we quantize activations and weights and validated on ImageNet, CIFAR-100, SQuAD and MNIST datasets. The results indicate that LLC achieves stable, lossless quantization across various models while maintaining high compression rates. On VGG series models, we even employed 2-bit quantization, as some layers were less sensitive to noise, and the INT2 noise boundary still fell within LLC's differential neighborhood on Cifar. In NLP tasks, such as question-answering with BERT, LLC compression continued to show

*Table 3.* Comparison of LLC quantization with existing methods while ensuring the same compression rate on ImageNet.

| Method | Top-1 | Top-5 | Loss | Drop-rate |
|---|---|---|---|---|
| Orgin(R.18) | 69.77 | 89.07 | 1.2470 | |
| AdaRound | 68.55 | - | - | |
| HAWQ | 69.56 | 88.97 | 1.2544 | ↓73% |
| ACIQ | 69.63 | 89.01 | 1.2492 | |
| Ours | **69.75** | **89.09** | **1.2457** | |
| Orgin(Mo_v2) | 71.89 | 90.29 | 1.1480 | |
| HAWQ | **72.90** | **90.97** | 1.1703 | |
| AdaRound | 69.25 | - | - | ↓70% |
| HAQ | 71.85 | 90.24 | - | |
| BRECQ | 72.57 | 90.24 | 1.1956 | |
| Ours | 71.89 | 90.30 | **1.1478** | |

*Table 4.* Performance of different models after decomposition on Imagenet. LLC steadily reduces the loss of the decomposed model.

| Model | Top-1 | Top-5 | Loss | Drop-rate |
|---|---|---|---|---|
| **Shallow Models** | | | | |
| VGG13_BN | **71.59** | 90.37 | 1.144342 | |
| Ours | 71.58 | **90.37** | **1.139801** | ↓39% |
| VGG19_BN | 74.21 | 91.84 | 1.042591 | |
| Ours | **74.22** | **91.89** | **1.021449** | ↓43% |
| ResNet-18 | **69.76** | **89.08** | 1.247314 | |
| Ours | 69.23 | 88.94 | **1.245241** | ↓62% |
| ResNet-50 | **76.13** | 92.86 | 0.961835 | |
| Ours | 76.10 | **92.90** | **0.950493** | ↓56% |
| **Deep Models** | | | | |
| ResNext101 | **79.31** | **94.52** | 0.926616 | |
| Ours | 78.16 | 94.02 | **0.869111** | ↓81% |
| ResNet-152 | **78.31** | 94.04 | 0.876225 | |
| Ours | 78.18 | **94.06** | **0.852449** | ↓10% |
| DenseNet169 | **75.60** | **92.81** | 0.997792 | |
| Ours | 75.45 | 92.80 | **0.971887** | ↓52% |

strong performance. Importantly, our focus is on stable, lossless compression rather than striving for lower-bit compression. Additionally, within the bounds of differential analysis, when compression noise opposes the gradient direction and has a larger magnitude (i.e., lower compression), the model loss decreases more substantially.

**Comparisons in Quantization.** Table 3 compares LLC with various quantization methods (Nagel et al., 2020; Dong et al., 2019; Banner et al., 2018; Wang et al., 2019; Li et al., 2021) at the same compression ratio. Existing methods generally lead to increased loss during quantization, whereas LLC achieves stable, lossless quantization through differential analysis. Although HAWQ slightly improves accuracy on MobileNet, it still incurs higher loss and fails to show consistent accuracy gains on other models. In contrast, LLC demonstrates stable performance across different models, effectively reducing model loss while maintaining broad applicability.

HAWQ series methods require multiple GPUs for quantization bit-width search, yet still take 30-50 minutes. In contrast, thanks to our efficient grouped knapsack search algorithm, our approach completes bit-width search in under 10 minutes on a single GPU. Additionally, since LLC's

*Table 5.* Comparison of LLC and existing methods on NLP datasets with the same compression ratio.

| SQuAD | Acc on Val | EM | F1 | Loss |
|---|---|---|---|---|
| BERT_base | 85.74 | 80.49 | 88.15 | 0.4461 |
| Base_SVD | 83.78 | 79.04 | 86.86 | 0.5168 |
| Zhang et.al | 84.33 | **80.48** | 87.94 | 0.6777 |
| Ours | **85.67** | 80.42 | **88.16** | **0.4460** |

| MNLI | Acc on Val | Loss on Val | Acc on Test | Loss on Test |
|---|---|---|---|---|
| BERT_base | 82.77 | 0.0289 | 83.91 | 0.0285 |
| Base_SVD | 81.69 | 0.0302 | 82.65 | 0.0299 |
| Song et.al | 81.46 | 0.0340 | 81.54 | 0.0310 |
| Zhang et.al | 80.54 | 0.0570 | 80.89 | 0.0742 |
| Ours | **82.78** | **0.0289** | 83.92 | **0.0285** |

lossless quantization process requires no fine-tuning or re-training, the quantization speed is extremely fast, taking less than 5 minutes in total. This demonstrates a significant efficiency advantage.

**Lossless in Decomposition.** In lossless decomposition, network depth significantly impacts model performance and matrix rank. Based on this, we divide models into shallow and deep categories for experiments, decomposing the linear layers on the ImageNet dataset. Table 4 presents the results of applying LLC to shallow and deep models. The results indicate that LLC enables lossless decomposition across different model architectures. Unlike quantization, decomposition alters the structure of the original parameter matrix, making compression more challenging. Nevertheless, LLC achieves reduced model loss while maintaining compression rates, demonstrating the effectiveness of its first-order differential analysis. Additionally, LLC shows lower loss than the original model and achieves comparable or even higher accuracy in some cases. It is noteworthy that LLC achieves a improvement in model loss reduction. This improvement is calculated using $\frac{\partial \ell}{\partial w_i} \cdot \delta_i^k$. Since both the gradient and compression noise values are less than 1, the extent of loss reduction cannot theoretically exceed 1.

**Comparisons in Decomposition.** Table 5 and 6 shows the performance of our proposed LLC method compared to other existing approaches (Zhang et al., 2023; Wei, 2021) on NLP datasets. Unlike current methods, LLC reliably achieves lossless model decomposition while significantly reducing model loss after compression. All methods in Table 5 use the same compression rate. We compressed the BERT model by 20% while maintaining leading accuracy. LLC consistently achieved loss reduction on both the validation and test sets, further demonstrating the generality and stability of the LLC decomposition approach.

Table 6 presents the efficiency and performance of LLC during the decomposition process. LLC achieves the shortest decomposition time and the lowest data requirements under the same hardware conditions. Existing low-rank decomposition methods often require fine-tuning and retraining to recover accuracy degradation, whereas our method reaches near-original model performance without the need for fine-

*Table 6.* The performance and efficiency of LLC compared to the existing methods. The efficiency of LLC decomposition is higher.

| ImageNet | Acc@5 | Cost Time(min) | Loss | Appor Data(G) |
|---|---|---|---|---|
| Orgin(VGG16) | 90.37 | - | 1.1443 | |
| SVD | 90.36 | 314.985 | 1.1454 | 140 |
| Tai et.al | 90.31 | - | - | 140 |
| Kim et.al | 89.4 | - | - | 140 |
| Zhang et.al | 90.35 | 433.115 | 1.2330 | 140 |
| Ours | **90.38** | **8.287** | **1.1393** | **6.4** |

| SWAG | Acc | Cost Time(min) | Loss | Appor Data(M) |
|---|---|---|---|---|
| Orgin(BERT) | **79.11** | - | 0.0579 | |
| SVD_ft | 78.44 | 134.649 | 0.0591 | 27 |
| Song et.al_ft | 78.55 | 151.006 | 0.0640 | 27 |
| Zhang et.al_ft | 79.00 | 233.146 | 0.0722 | 27 |
| Ours | 78.57 | **11.413** | **0.0566** | **7.6** |

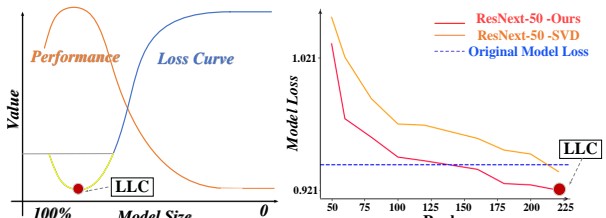

*Figure 2.* Performance curves and loss curves of LLC in quantization and decomposition methods. LLC can achieve better performance with lower loss and smaller models.

tuning or retraining, outperforming most existing methods.

Figure 2 illustrates the performance and loss curves for LLC when compressing the ResNext-50 model. During LLC quantization, LLC automatically selects the optimal bit-width for lossless compression, while during decomposition, it identifies the lowest rank suitable for lossless compression. Compared to SVD methods, LLC more reliably identifies low-rank matrices that preserve accuracy, achieving effective model compression.

**Discussion and Limitations.** The core principle of LLC is to leverage total differentiation to establish an error neighborhood, identifying compression vectors that oppose the gradient direction to ensure the compressed model has lower loss than the original. Thus, LLC aims for stable, lossless compression rather than maximizing compression ratio. When the quantization bit-width and rank are extremely low, the resulting error margin expands beyond the scope of low-order total differential analysis, making theoretically lossless compression unfeasible.

# 6. Conclusion

This paper introduces a general loss-driven lossless compression framework designed to achieve stable and lossless model compression. LLC defines the compression neighborhood and higher-order analysis boundaries through total differentiation, specifying the permissible error range for lossless model compression. Ultimately, LLC has been effectively applied to both quantization and decomposition, achieving efficient compression outcomes.

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
