# OpenReview forum: "Compression for Better: A General and Loss-Driven Compression Framework"
_ICML.cc/2025/Conference — Submitted to ICML 2025_

### Official Review · Reviewer_rFRy · 2025-03-08

**Overall Recommendation:** 2

**Summary:**

This paper introduces LLC, a LossLess model Compression framework that specifies the permissible error range for lossless model compression through higher-order error bound analysis. LLC is applied to both quantization and decomposition, achieving notable compression results without compromising performance.

**Update After Rebuttal**: In summary, ensuring lossless compression requires that Eq. 7 remains negative and that a solution exists at each layer. The activation value is influenced by input, while the sign of each parameter is determined post-quantization. However, the paper does not address how the existence of a solution is guaranteed.
Additionally, the interaction between layers is captured by the second-order Hessian. While the gradient may dominate, this does not imply that the off-diagonal elements of the Hessian have no effect on the results. Prior works, such as BRECQ and QDrop, have explored this aspect, and I do not see a significant theoretical distinction in this paper.
After considering the feedback from other reviewers, I have decided to maintain my score.

**Claims And Evidence:**

The paper’s claims are supported by experimental results. However, further theoretical validation would strengthen the argument.

**Essential References Not Discussed:**

None.

**Experimental Designs Or Analyses:**

i.	Are the calibration dataset and test dataset different? How does the calibration dataset compare to those used in other baseline methods?

ii.	Does "drop-rate" refer to the compression ratio of model weights?

**Methods And Evaluation Criteria:**

The proposed method is interesting, especially in considering the direction of compression error.

**Other Comments Or Suggestions:**

The theoretical foundation of the paper is not entirely rigorous (see theoretical claims). With layer-wise error accumulation, it is unclear whether a lossless compression solution always exists. Specifically, since quantization affects subsequent layers, its joint influence should be considered. However, Equation (3) and subsequent analysis are based on individual sum of error, do not seem to account for this. Moreover, this work presents counterintuitive results. The authors may provide additional details or consider open-sourcing their implementation.

**Other Strengths And Weaknesses:**

None

**Questions For Authors:**

a)	The authors use gradient-based criteria. In the case of quantization, does 2-bit quantization always satisfy the error bound based only on the gradient, as shown in Figure 1? If not, how is lossless compression ensured?

b)	What are scale_input and scale_weight in Algorithm 1? Do they refer to the scaling factors used for floating-point to integer conversion? What is the exact role of noise in Algorithm 1? The description of the algorithm should be corrected and clarified.

c)	The authors should specify the layer-wise bit-width and rank configurations for lossless compression. Within a given layer, do all weights share the same bit-width?

**Relation To Broader Scientific Literature:**

This work is relevant to the field of model compression.

**Theoretical Claims:**

I have reviewed the theoretical derivation and found some issues. In particular, I question whether each layer, such as a convolutional layer, can always be represented as \frac{\partial l_k}{\partial h_i} \cdot \delta_i, as assumed in the paper. This directly affects the correctness of the scalar product formulation in Equations (7) and (8), as well as the implementation in the algorithm. The authors may need to provide additional mathematical justification for this step.

---

> ### Author Rebuttal · Authors · 2025-04-01
>
> Q1: About convolution representation.
>
> A1:
> 1. The convolution layer can be expressed in the first-order differential form[2]. This first-order approximation is applicable to any differentiable operation unit (including conv layers, etc.), and its effectiveness comes from the differentiability assumption of neural networks[1]. The following is a simple proof,
>
> Assume that the output $h_i$ of a conv layer is defined by the convolution:
> $
> h_i = (w * x)_i =\sum_j w[i - j]x_j
> $
>
> where $x$ is the network input, $w[i - j]$ represents the convolution kernel weights, and we introduce a new index $j$ to denote the input position corresponding to kernel computation. $h_i$ is the activation at position $i$. Suppose the input $x$ is perturbed by a small amount $\delta$, i.e., $x \rightarrow x + \delta$.
>
> Then, the new convolution output becomes:
> $
> \tilde{h}_i = (w * (x+\delta))_i = \sum_j w[i - j]\Bigl(x_j + \delta_j\Bigr)
> $
>
> Due to the linearity of convolution, we can split the above sum:
> $
> \tilde{h}_i = \sum_j w[i - j]x_j +\sum_j w[i - j]\delta_j = h_i + (w * \delta)_i
> $
>
> Thus, the perturbation is:
> $
> \Delta h_i = \tilde{h}_i - h_i = (w * \delta)_i
> $
>
> To be consistent with the notation in the paper, we denote:
> $
> \delta_i = (w * \delta)_i
> $
>
> This means that the input perturbation $\delta$ is weighted by the kernel $w$ to yield the output perturbation $\delta_i$. Assuming the loss function $l_k$ is differentiable with respect to the activation $h_i$, a first-order expansion gives:
> $
> l_k(h_i + \Delta h_i) \approx l_k(h_i) + \frac{\partial l_k}{\partial h_i}\Delta h_i
> $
>
> Replacing $\Delta h_i$ with $\delta_i$ yields:
> $
> \Delta l_k \approx \frac{\partial l_k}{\partial h_i} \cdot \delta_i
> $
>
> For a detailed derivation and differentiation, please refer to https://www.jefkine.com/general/2016/09/05/backpropagation-in-convolutional-neural-networks/.
>
> *[1]Goodfellow I et al. Deep learning[M]. Cambridge: MIT press, 2016.*
>
> *[2]Dumoulin V. A guide to convolution arithmetic for deep learning[J]. arXiv preprint 2016.*
>
> Q2: About accumulation joint impact.
>
> A2: The accumulation of layer-wise errors comes from the off-diagonal elements of the Hessian matrix in the second-order terms. Through extensive experimentation, in the first-order neighborhood, we have found that the impact of the second-order terms on the loss is smaller than $10^{-5}$, and the layer-wise accumulated error is similarly smaller than $10^{-5}$.
>
> Because the computation of second-order terms is time-consuming, LLC neglects these terms. The joint impact and the summation of multiple errors that you referred to are typically results when the second-order terms dominate, which is different from LLC’s approach.
>
> Q3: About the calibration and drop-rate.
>
> A3: The calibration dataset is completely different from the test dataset. We follow the existing mixed precision quantization work and keep the calibration set consistent with the existing methods[Zhang et.al]. Drop-rate refers to the compression rate of the model weights.
>
> Q4: About the error boundary.
>
> A4:  After extensive experiments, we have found it is less common for layers quantized to 2 bits. 2-bit quantization does not always satisfy the gradient-based error bound. As shown in Fig 1, when the error caused by ultra-low bit is large, the first-order term still dominates, but the error introduced by the second-order term is non-negligible ($10^{-2}$), it may not ensure lossless compression. This issue has been addressed in the LLC limitation section. Therefore, in such cases, it is necessary to additionally analyze the impact of the high-order term on the model, which is also the next research direction of LLC.
>
> Since the purpose of LLC is to ensure losslessness and select a suitable bit width for the layer, if LLC cannot ensure lossless compression under ultra-low bits, it usually increases the bit width (2bit->4bit) to minimize the loss. The experimental results are detailed in Reviewer 7rwL A2.
>
> Q5:About noise.
>
> A5:$scale_{input}$ and $scale_{weight}$ are the scaling factors used for the FP to Int conversion. In Alg 1, the role of the noise is twofold:
>
> (a)During the calibration process, the noise is used to compute the magnitude of quantization errors to ensure that they remain within the differential neighborhood;
>
> (b)During the quantization process, the noise represents the quantization error and must remain opposite to the gradient direction.
>
> We have updated the algorithm and clarified the symbols.
>
> Q6: About configuration.
>
> A6: LLC adopts a layer-wise quantization approach, where all weights in a given layer use the same bit-width. This is consistent with existing work. For specific configurations of bit-width and rank, we plan to gradually release the code to provide more information.
>
> We have updated the paper as you suggested. However, due to ICML's rebuttal policy, the updated paper will be presented later. I hope the reply can address your concerns and look forward to your feedback.

---

### Official Review · Reviewer_645a · 2025-03-11

**Overall Recommendation:** 2

**Summary:**

This paper proposes a general theoretical framework to achieve lossless compression. The paper uses a loss-driven framework to specify the error range each layer's weight and activation can tolerate. A model compression scheme such as quantization or decomposition can therefore be searched within the error ranges.

## update after rebuttal

After checking other reviews and the author's reply, I still think the theortical framework proposed in this work is overly simplified when dealing with piecewise linear activation functions and dealing with cross-layer correlations. The experimental results are also weak in supporting the effectiveness of the proposed work. I retain my score of weak rejection.

**Claims And Evidence:**

The derivation of the paper is based on the assumption that the function of the neural network is inherently smooth and differentiable, yet as most models are using ReLU activations, it is unsure if the smoothness condition still holds given the piece-wise nature of the ReLU function.

In the noise neighborhood mapping, the treatment in Equ. (5) appears to consider the perturbations in different layers are independent. As this holds for weight perturbation, activation perturbation is correlated to the perturbations in earlier layers. It is not sure how this correlation is taken into the account.

**Essential References Not Discussed:**

Reference adequate

**Experimental Designs Or Analyses:**

It is unclear which part of the dataset is used for calibrating the noise bound and which part is used for reporting the experiment results. It is mentioned in Sec. 5.1 that "the validation set" is used for calibration, which is a doubtful choice. If the validation set is used for both noise bound calibration and performance evaluation, the proposed method may subject to overfit, where the noise bound is ony effective on the calibration dataset but not generalize.

Additionally, the noise bound computation and the optimization within the bound is complicated and may be costly. The cost is only reported for SVD experiments but not for quantization experiments.

**Methods And Evaluation Criteria:**

Loss is reported as the main criteria in the experiments. This makes sense given the target of the propose dmethod is to achieve a lossless compression. However, it appears that a smaller loss does not always lead to a better performance, and the performance improvement over baseline methods is limited. This implies that the objective of controlling loss difference may not be practically useful.

**Other Comments Or Suggestions:**

No other comments

**Other Strengths And Weaknesses:**

Besides what is mentioned previously, the data source of Fig. 1 is also unclear. It is not clear how the noise boundaries in Fig. 1 are computed and what is it's impact on the method.

**Questions For Authors:**

Is the proposed method overfitting to the calibration dataset?

**Relation To Broader Scientific Literature:**

This paper follows the line of work that use Taylor Expansion based criteria to estimate the impact of model compression noise to the model loss. Unlike previous work that tries to minimize the loss differnet, this work derives a noise boundary that guarantee a smaller loss. This prespective is novel and not explored before.

**Theoretical Claims:**

Given the assumptions made in the paper, the theoretical derivation of the LLC framework appears to be correct.

---

> ### Author Rebuttal · Authors · 2025-03-31
>
> Q1: About Smoothness Condition.
>
> A1: Except at the zero point, ReLU is smooth and differentiable in most regions; its non-differentiability occurs only when the activation value is exactly zero. Since real-world datasets (e.g., ImageNet) typically feature continuous distributions, the activations input to ReLU are also mostly continuous real numbers. This implies that the probability of activations being exactly zero is extremely low, and thus the piecewise nature of ReLU has a negligible impact on the overall optimization process. Therefore, ReLU does not invalidate the overall smoothness assumption, and the theoretical derivation of LLC remains valid.
>
> *Bengio, Y. On the Number of Linear Regions of Deep Neural Networks. [NIPS2014]*
>
> Q2: About the correlation of layers.
>
> A2: In Equation 5, we have accounted for the inter-layer perturbation correlations, mathematically manifested as the effect of the off-diagonal elements of the second-order Hessian matrix (denoted as $H_{i,j} = \frac{\partial^2 f}{\partial x_i \partial x_j}$). This term reveals some dependency among layers.
>
> Q3: About limited loss improvement.
>
> A3:
> 1. First of all, from the perspective of the improvement, the loss improvement is competitive. For example, in ResNet18, as shown in the following table, the data in the table represents the improvement compared with the original model after compression, and the larger the value, the better. HAWQ has a smaller improvement over LLC. The improvement compared with other methods is also competitive.
> ResNet18|Acc. Drop$\uparrow$|Loss. Drop$\uparrow$
> -|-|-
> HAWQ|-0.21%|-0.74%
> ACIQ|-0.14%|-0.22%
> LLC|**-0.02%**| **+0.13%**
>
> 2. You mentioned that smaller loss does not always seem to lead to better performance. This is because you have already believed that loss (cross entropy, sequence cross entropy, AUC error, etc.) cannot represent the performance of the model. But in fact, cross entropy or accuracy are all performance indicators that can evaluate the model.
>
> You agree with LLC's theoretical framework for loss. As you mentioned, loss is the main criterion in the experiment. Therefore, in LLC's experiments, loss (cross entropy, sequence cross entropy, AUC error, etc.) is mainly used to measure the performance of the model in different tasks. For more discussion on loss, please refer to the answer to m1Ar reviewer. Looking forward to your discussion with us.
>
> Q4: About calibration set.
>
> A4: Following existing pipelines, we use the ImageNet validation set to assess the noise boundary and the test set for performance evaluation. Since the validation and test sets contain entirely different data, the calibration set is used solely for computing quantization parameters and error boundaries, thus avoiding overfitting to the calibration data. All experimental results in the paper are based on the test set.
>
> Q5: About the noise boundary.
>
> A5: The data source of Figure 1 is calculated by Eq. 6. In the early stage, we conducted a lot of experiments to calculate the compression boundary. That is, control the size of the introduced noise, measure the applicable range of the differential expansion, and observe the change in loss. The purpose is to analyze the relationship between the introduced noise and the higher-order terms. For example, when the error introduced by the weight is less than 8e-3, the first-order gradient is the main factor affecting the compression performance. Specifically, in this experiment, we mainly perturb the ResNet18, ResNet34, ResNet50, Bert and other models with different error sizes. After the perturbation, different error ranges for the first-order and second-order terms are obtained.
>
> The calculation of the noise boundary of the second-order term is complicated. We use the Lanzcos algorithm to restore the second-order Hessian term as much as possible. The calculation time takes several days and the cost is high. However, after repeated experiments, we found that the impact of the second-order term is small within the range of $8\times10^{-3}$, and the time cost is large. Therefore, LLC ignores the calculation of the second-order term and focuses on the first-order term. At the same time, LLC calculates the cost of the noise boundary of the first-order term is extremely small, which takes only about 3-4 minutes. The experimental hardware uses NVIDIA A800 GPU.
>
> We show the cost of the quantization experiment in the table below. LLC has higher efficiency in both quantization and decomposition.
> Model |ResNet18|ResNet34| ResNet50|BERT
> -|-|-|-|-
> Time(min)|8.42|12.66|14.99|17.34
>
> We have added the above experimental results to the original text and added a description of the noise boundary in Figure 1 based on your comments. Due to the rebuttal policy of ICML, we are currently unable to display the original paper publicly. This paper will be provided in subsequent stages.
>
> Thank you for your suggestions and hope that our answer can address your concerns. Looking forward to your feedback.

---

### Official Review · Reviewer_7rwL · 2025-03-11

**Overall Recommendation:** 2

**Summary:**

This paper presents a novel theoretical framework, LossLess Compression (LLC), which provides a principled approach to model compression while ensuring that the model’s loss remains unchanged or even decreases after compression. Through extensive experimentation, LLC demonstrates its effectiveness in achieving lossless model compression across various architectures and datasets, achieving compression ratios of up to 70% while maintaining or improving model performance.

**Update after rebuttal**:
The author only solved part of my confusion. The authos fails to demonstrate its effectiveness in diffusion models and large multimodal models. I still maintain my decision to weak reject.

**Claims And Evidence:**

Claim: "LLC can achieve compression without performance degradation across all models and datasets".
The claim that LLC universally prevents degradation is not fully proven: some results (e.g., Table 4, ResNet-18) show minor drops in accuracy after compression. The paper does not explore cases where compression might exceed the lossless threshold, potentially leading to degradation. In addition, the authors does not apply their method to LLM and diffusion models.

Claim: "LLC can always find an optimal compression ratio that reduces loss". This claim that loss after compression is always lower than before is too strong. While some cases show lower loss post-compression, this is not universal across all models and datasets.
The mathematical formulation relies on first-order analysis, which may not fully capture second-order or higher-order compression effects, particularly in extreme compression scenarios.

**Essential References Not Discussed:**

The paper compares against HAWQ (ICCV'19) but ignores HAWQ-V3 [ICML'21], which further improved Hessian-based bit-width selection.

**Experimental Designs Or Analyses:**

No extreme compression analysis (e.g., 1-bit quantization, rank-1 decomposition). LLC assumes first-order approximations are sufficient, but no tests explore when this breaks down (e.g., 1-bit quantization or extreme low-rank factorization).

No layer-wise sensitivity analysis. Different layers in deep networks tolerate compression differently, but the paper does not analyze which layers benefit most from LLC.

Limited model diversity. The paper does not evaluate LLM and diffusion models. These models behave differently than CNNs/RNNs, and LLC may not generalize to them.

**Methods And Evaluation Criteria:**

1. Need for more diverse models. LLC is claimed to be a general framework, but testing on only a handful of architectures does not guarantee universal applicability. It is necessary to test on LLM and diffusion models.

2. The framework assumes first-order approximations are sufficient, but extreme compression (e.g., 1-bit quantization or ultra-low-rank decomposition) might violate these assumptions.

**Other Comments Or Suggestions:**

No

**Other Strengths And Weaknesses:**

Strengths:
1. The paper introduces a novel formulation for compression via total differential analysis, which explicitly models the effect of compression-induced perturbations on the loss function.

2. Unlike many heuristic-based quantization and decomposition methods, LLC derives explicit mathematical conditions under which compression can be performed without increasing loss.

3. Reformulating quantization bit-width selection as a grouped knapsack problem is a novel idea that improves efficiency compared to brute-force search approaches (e.g., HAWQ’s bit-width exploration).

Weakness:
1. Overclaimed about universal loss reduction. The paper does not analyze when LLC fails to maintain loss, making the "universal loss reduction" claim too strong.

2. Lack of analysis for extreme compression scenarios. LLC assumes that first-order approximations are sufficient, but this may not hold for aggressive quantization (e.g., 1-bit, 2-bit). The paper does not test LLC at very low bit-widths, making it unclear whether LLC remains lossless under extreme compression.

3. The competing baselines are proposed in several years ago. The authors should compare their method with the latest baselines.

**Questions For Authors:**

No

**Relation To Broader Scientific Literature:**

1. Reformulate quantization as a grouped knapsack problem, which reduces search complexity compared to heuristic-based approaches like HAWQ. Unlike BRECQ, LLC does not require fine-tuning, making it computationally cheaper.

2. Extend loss-sensitive compression methods by providing an explicit mathematical formulation (via total differentials) to define the safe compression region. Unlike HAWQ, LLC does not require second-order Hessian computation, making it computationally cheaper.

3. Provide an explicit compression boundary formulation, improving interpretability compared to prior empirical loss landscape studies, such as Parr & Howard (2018), Ghorbani et al. (2019), Li et al. (2018).

**Theoretical Claims:**

The paper formulates model loss as a function of compression noise using total differential, which neglects the second-order terms. It assumes that the Hessian contributions are always negligible, which may not hold in cases where compression induces significant changes.

The bit-width selection problem in mixed-precision quantization can be efficiently solved as a grouped knapsack problem. However, the knapsack formulation assumes that bit-width choices per layer are independent, but in reality, inter-layer dependencies exist (e.g., some layers are more robust to quantization than others).

---

> ### Author Rebuttal · Authors · 2025-03-31
>
> Q1:About when LLC cannot reduce loss
>
> A1:Regarding the limitations of LLC, we emphasize that the loss reduction achieved by LLC is not exaggerated but based on strictly defined mathematical conditions. This sentence has been mentioned in the advantages of the paper you commented on: LLC has clear mathematical conditions.
>
> As described in the paper, LLC can reduce the loss of the model in the differential neighborhood; **in section 3**, we discussed the effective conditions in the noise neighborhood mapping, which require that the perturbation is small enough to ensure the effectiveness of the total differential expansion. Only in such a differential neighborhood can LLC guarantee the reduction of loss. Once the perturbation exceeds this neighborhood, the influence of high-order terms becomes significant, and LLC can no longer guarantee the reduction of loss, making the total differential analysis no longer applicable, and further analysis of high-order terms is required. Therefore, the effective conditions of LLC are the effective conditions of the total differential expansion.
> When the significant error caused by extreme compression is introduced, the total differential analysis is difficult to apply, and LLC cannot maintain the loss, which is also mentioned in **the limitations of the paper**.
>
> Q2:About extreme compression and sensitivity analysis
>
> A2:Our algorithm is still based on mixed-precision compression, and we use quantization as an example for explanation. First, the purpose of LLC is to find the bit width that can reduce the model loss, so the bit width selected is different according to the tolerance of the layer. LLC is best-effort, that is, when the layer can be extremely compressed (meeting the neighborhood condition), LLC will compress it to ultra-low bits. Taking VGG13 as an example, the following table shows the loss changes of LLC for quantization of the 4th and 8th layers,
>  VGG13(*L*:1.2726) |8bit|4bit|2bit|1bit
> -|-|-|-|-|
> *L*(#8) | 1.2689 | 1.2603 | **1.2599** | 2.1599
>  *L*(#4) | 1.2711 | **1.2709** | 1.9961 | 2.6634
>
> The error caused by 1bit significantly exceeds the neighborhood range of these two layers, so 1bit compression cannot be performed. LLC will increase the bit width to 2bit compression. At this time, the eighth layer meets the neighborhood condition and the loss is reduced. However, the fourth layer still exceeds the neighborhood range, resulting in increased loss. Therefore, 4bit compression is finally selected for this layer. This example demonstrates that layers with larger error neighborhoods benefit most from LLC.
>
> Secondly, LLC defines "aggressive compression" based on the impact of low-bit errors on the model's loss. Since different layers have varying tolerances to compression, when the error introduced by 2-bit quantization remains within the first-order differentiable neighborhood, LLC can achieve loss reduction with 2-bit compression. As shown in Table 2 of the original paper, some models maintain lossless performance even after 2-bit compression.
>
> Q3: About inter-layer dependency
>
> A3: Mathematically, the inter-layer dependency primarily stems from the off-diagonal elements $H_{i,j} = \frac{\partial^2 f}{\partial w_i \partial w_j}$ of the second-order Hessian matrix. However, experimental results indicate that within the first-order neighborhood, the contribution of the second-order terms is negligible, and the first-order terms remain dominant. Consequently, within this neighborhood, LLC can reasonably neglect inter-layer dependencies, effectively transforming the bit-width selection problem in mixed-precision quantization into an approximately independent grouped knapsack problem.
>
> Q4: About comparison
>
> A4: The following table is a comparison with the latest baseline method. LLC is competitive and the accuracy drop is still minimal.
>
>  ResNet50 | Orgin Acc. | Quant Acc.  | Error | Model Size
> -|-|-|-|-
>  HAWQV3[ICML'21] | 77.72 | 77.58 | -0.14 | 24.5
>  Qdrop[ICLR'22] | 76.8 | 76.65 | -0.15 | 24.5
>  PTMQ[AAAI'24] | 76.8 | 76.52 | -0.28 | 24.5
>  LLC | 75.06 | 75.04 | **-0.02** | 24.5
>
> The following table shows LLC's performance on Tinyllama compression on MMLU. LLC still improves accuracy while reducing loss. Due to rebuttal policy we were unable to submit the latest paper, but more results and details have been added to the original paper.
>
>  LLM| STEM | Hum. | Social. | Other | Acc. | SFT Loss
> -|-|-|-|-|-|-
>  Orgin | 26.938 | 24.378 | 30.971 | 26.681 | 26.904 | 1.769011
>  LLC | **27.469** | **24.846** | **31.199** | **26.959** | **27.29** | **1.763389**
>
> Due to our lack of familiarity with the neighborhood characteristics of the diffusion model, coupled with the limitations of rebuttal time and calibration datasets, it is currently difficult to quantify and analyze the diffusion model within a limited time. We plan to explore this direction in future work, and hope you understand.
>
> Thank you for your suggestions and we look forward to receiving your feedback.

---

### Official Review · Reviewer_m1Ar · 2025-03-24

**Overall Recommendation:** 2

**Summary:**

The paper proposes a general model compression framework named LossLess Compression theoretical framework(LLC), which focuses on reducing the model loss for better model performance. By considering quantization and decomposition as adding noise to the model weights and activations, the loss introduced by model compression on the whole dataset can be represented. With the analytic experiments, the authors find that the first-order term of the Hessian matrix has dominant influence on the loss. Therefore, the higher-order terms of the Hessian matrix is omitted for efficiency in the LLC framework. The authors evaluate the performance of LLC with model quantization and decomposition on both computer vision and natural language processing datasets. As shown in the results, the models compressed with LLC mostly achieves lower loss values compared with the baseline models.

**Claims And Evidence:**

Yes. The results show that LLC with quantization and decomposition achieve lossless compression as claimed on different models and datasets.

**Essential References Not Discussed:**

None.

**Experimental Designs Or Analyses:**

The results show that LLC with quantization and decomposition achieve lossless compression as claimed on different models and datasets. While it doesn't show outstanding performance in the comparison experiments as shown in Table 3 and 6. In the Table 6, the MobileNet-V2 with LLC achieves the accuracy of 71.89%. It is lower than that of HAWQ (72.90%), which is published in 2019. There should be more comparison results with the state-of-the-art compression methods.

**Methods And Evaluation Criteria:**

Whether the proposed methods make sense depends on the mathematic proofs. I'm not a math expert. Please refer to other's comment.
The evaluation criteria including the datasets, models and metrics are reasonable for me.

**Other Comments Or Suggestions:**

None.

**Other Strengths And Weaknesses:**

Strengths:
1.The paper provides pseudo code to describe the methods.
2.The time cost of the proposed compression method is relatively low.

Weaknesses:
1.There is no illustration about the proposed method to help the reader understand.
2.Inconsistent text format. (page 5)

**Questions For Authors:**

1. As shown in the Table 3, LLC achieves lower loss values but lower accuracy on mobilenet-v2 compared with HAWQ. How to explain the phenomenon?
2. It shows that lower loss value doesn't indicate higher accuracy. Do you think the loss value is a good evaluation metric? Why?

**Relation To Broader Scientific Literature:**

The proposed LLC framework aims to achieve lossless compression. With LLC, the compressed model can even achieve lower loss value compared with the full-precision model. It shows that compression not always leads to degradation, which is the key contribution of the paper.

**Theoretical Claims:**

The mathematical proofs for Equation 1,2,...,6 seems reasonable to me. For the remaining proofs, which corresponds to the core design of LLC, please refer to other's comment.

---

> ### Author Rebuttal · Authors · 2025-03-31
>
> Q1: About the evaluation metrics in Table 3.
>
> A1:
> First, in Table 3, the decrease in loss value represents the decrease in cross entropy. Cross entropy and accuracy represent different evaluation methods. Compared with accuracy, cross entropy can compare the closeness between the probability distribution predicted by the model and the true distribution. That is to say, cross entropy can more carefully reflect the performance of the model on different samples, not only whether it is correct or not, but also the degree of certainty of the prediction. At the same time, accuracy will be dominated by the performance of most categories. Compared with HAWQ, although LLC has a slightly lower accuracy, LLC performs better than HAWQ under the evaluation standard of cross entropy. This shows that lower loss values can reveal more detailed improvements in the probability distribution of the model. Secondly, compared with the original model Mobilenet-v2, LLC not only outperforms the original model in accuracy, but also has a lower cross entropy than the original model. HAWQ outperforms the original model only in the accuracy evaluation metric, but is lower than the original model in the cross entropy evaluation metric.
>
> Q2: About loss and accuracy.
>
> A2:
> 1. Different focus. Loss functions remain a valid evaluation metric, such as cross-entropy, sequence cross-entropy, and AUC error, among others. We believe that both cross-entropy and accuracy reflect model performance, regardless of good or bad, but they emphasize different aspects. For our work, as described in Q1, loss functions like cross-entropy are better suitable, making them a more appropriate evaluation standard.
>
> 2. Generality metric. As mentioned in the paper, we aim to develop an evaluation method that is applicable to different tasks and technical frameworks to validate the universality of LLC. Since different tasks or models may employ diverse evaluation metrics, and the loss function, as an inherent property of the model, can uniformly reflect the model’s performance across various tasks, using loss as the evaluation metric is the best choice for verifying the universality of LLC.
>
> 3. LLC does not target specific forms of loss, but it can steadily reduce losses. LLC is a compression method for loss reduction, that is, compression may lead to performance improvement. In order to prove the versatility of LLC, LLC has been verified on different tasks, data sets, and compression methods. More importantly, there are differences in the loss indicators used in different tasks in this paper, but LLC can achieve the reduction of loss after compression, proving that it is independent of the loss function. Theoretical analysis and experimental results can show that, under the differential framework, LLC implements a method for reducing loss after compression for a variety of losses (cross entropy, sequence cross entropy, AUC error, etc.) and compression techniques (quantization and decomposition). In other words, LLC also provides an explanation for the performance improvement after compression, which is also used as a guide for future compression work.
> 4. Reasonableness of loss evaluation.
>  Regarding evaluation metrics, we believe that the loss value best reflects the effectiveness of LLC. Because the starting point of LLC is the analysis of model loss, the purpose is to reduce the loss values (cross entropy, sequence cross entropy, AUC error, etc.) on different tasks. It is undeniable that most papers with theoretical analysis start with loss, and theoretical analysis ultimately achieves the purpose of reducing loss. However, in the experiment, although these works may have advantages in other indicators, they avoid the evaluation of loss (such as HAWQ, etc.). So are there certain flaws in such theoretical verification? If these works lead to higher model loss, does it mean that there are certain flaws in the theoretical analysis and it does not match the experimental results? Based on the above concerns, we use loss to evaluate and verify the rationality of the theory, despite similar papers avoiding experiments on loss.
>
> Thank you for your valuable feedback. We are honored to discuss this issue with you and look forward to your further comments.

---

### Decision · Program_Chairs · 2025-05-01

**Decision:**

Reject

**Comment:**

This paper presents a theoretical framework that claims to achieve lossless compression of pre-trained models, from the perspective of ensuring the model loss to be unchanged or smaller after compression. The proposed approach simply relies on the first-order term of the Hessian matrix while neglecting the second-order terms. The submission was initially/finally scored (2,2,2,2) by three knowledgeable reviewers and another reviewer, who raised several major concerns about 1) unconvincing motivation and experimental support, smaller loss does not always indicate better accuracy; 2) limited technical novelty, especially over many previous methods like HAWQ family, BRECQ, QDrop, AdaRound and so on; 3) weak experiments, without using any current mainstream models especially LLMs, smaller loss leads to accuracy drop as observed in experiments, only mix-precision quantization settings (less popular in practice) are considered, experimental details are missing (calibration data, layer-wise bit-width settings on each dataset, etc.); 4) questionable core contribution.

The authors provided detailed responses to these concerns, and all reviewers acknowledged the authors' rebuttal, but pointed out that their main concerns were not resolved. The AC read the paper, the reviews, the rebuttal and the reviewers' feedback, and mostly agree with the reviewers' assessment. Generally, this paper at its current form does not meet the bar for acceptance, in terms of novelty, experimental design, and presentation. The authors are encouraged to consider the reviewers' comments and suggestions to improve their work for a future conference.